# Improving Practice for Urinary Continence Care on Adult Acute Medical and Rehabilitation Wards: A Multi-Site, Co-Created Implementation Study

**DOI:** 10.3390/healthcare11091241

**Published:** 2023-04-26

**Authors:** Dianne Lesley Marsden, Kerry Boyle, Jaclyn Birnie, Amanda Buzio, Joshua Dizon, Judith Dunne, Sandra Greensill, Kelvin Hill, Sandra Lever, Fiona Minett, Sally Ormond, Jodi Shipp, Jennifer Steel, Amanda Styles, John Wiggers, Dominique Ann-Michele Cadilhac, Jed Duff

**Affiliations:** 1Hunter Stroke Service, Hunter New England Local Health District, New Lambton Heights, NSW 2305, Australia; kerry.boyle@health.nsw.gov.au (K.B.); sally.ormond@health.nsw.gov.au (S.O.); 2College of Health, Medicine and Wellbeing, University of Newcastle, Callaghan, NSW 2308, Australia; john.wiggers@health.nsw.gov.au (J.W.); j2.duff@qut.edu.au (J.D.); 3Hunter Medical Research Institute, New Lambton Heights, NSW 2305, Australia; 4Belmont Hospital, Hunter New England Local Health District, New Lambton Heights, NSW 2305, Australia; 5Armidale Hospital, Hunter New England Local Health District, Armidale, NSW 2350, Australia; jaclyn.birnie@health.nsw.gov.au (J.B.); amanda.styles@health.nsw.gov.au (A.S.); 6Coffs Harbour Health Campus, Mid North Coast Local Health District, Coffs Harbour, NSW 2450, Australia; 7Rankin Park Centre, Hunter New England Local Health District, New Lambton Heights, NSW 2305, Australia; judith.dunne@health.nsw.gov.au (J.D.); jodi.shipp@health.nsw.gov.au (J.S.); 8John Hunter Hospital, Hunter New England Local Health District, New Lambton Heights, NSW 2305, Australia; 9Rockhampton Hospital, Central Queensland Hospital and Health Service, Rockhampton, QLD 4700, Australia; sandra.greensill@health.qld.gov.au; 10Stroke Foundation, Melbourne, VIC 3000, Australia; khill@strokefoundation.org.au; 11Ryde Hospital, Northern Sydney Local Health District, Eastwood, NSW 2122, Australia; 12Susan Wakil School of Nursing and Midwifery, The University of Sydney, Sydney, NSW 2006, Australia; 13Manning Hospital, Hunter New England Local Health District, Taree, NSW 2430, Australia; fiona.minett@health.nsw.gov.au; 14Wingham Hospital, Hunter New England Local Health District, Wingham, NSW 2429, Australia; 15Calvary Mater Newcastle, Waratah, NSW 2298, Australia; 16Port Macquarie Hospital, Mid North Coast Local Health District, Port Macquarie, NSW 2444, Australia; jennifer.steel@health.nsw.gov.au; 17Tamworth Hospital, Hunter New England Local Health District, Tamworth, NSW 2340, Australia; 18Health Research and Translation, Hunter New England Local Health District, New Lambton Heights, NSW 2305, Australia; 19Stroke and Ageing Research, Faculty of Medicine, Nursing and Health Sciences, School of Clinical Sciences at Monash Health, Clayton, VIC 3168, Australia; dominique.cadilhac@monash.edu; 20School of Nursing and Centre for Healthcare Transformation, Queensland University of Technology, Brisbane, QLD 4001, Australia; 21Royal Brisbane and Womens Hospital, Queensland Health, Brisbane, QLD 4029, Australia

**Keywords:** urinary incontinence, lower urinary tract symptoms, inpatient, patient care planning, professional practice gaps, evidence-based practice, nursing process, hospital, implementation science, quality improvement

## Abstract

Many adult inpatients experience urinary continence issues; however, we lack evidence on effective interventions for inpatient continence care. We conducted a before and after implementation study. We implemented our guideline-based intervention using strategies targeting identified barriers and evaluated the impact on urinary continence care provided by inpatient clinicians. Fifteen wards (acute = 3, rehabilitation = 7, acute and rehabilitation = 5) at 12 hospitals (metropolitan = 4, regional = 8) participated. We screened 2298 consecutive adult medical records for evidence of urinary continence symptoms over three 3-month periods: before implementation (T_0_: n = 849), after the 6-month implementation period (T_1_: n = 740), and after a 6-month maintenance period (T_2_: n = 709). The records of symptomatic inpatients were audited for continence assessment, diagnosis, and management plans. All wards contributed data at T_0_, and 11/15 wards contributed at T_1_ and T_2_ (dropouts due to COVID-19). Approximately 26% of stroke, 33% acute medical, and 50% of rehabilitation inpatients were symptomatic. The proportions of symptomatic patients (T_0_: n = 283, T_1_: n = 241, T_2_: n = 256) receiving recommended care were: assessment T_0_ = 38%, T_1_ = 63%, T_2_ = 68%; diagnosis T_0_ = 30%, T_1_ = 70%, T_2_ = 71%; management plan T_0_ = 7%, T_1_ = 24%, T_2_ = 24%. Overall, there were 4-fold increased odds for receiving assessments and management plans and 6-fold greater odds for diagnosis. These improvements were sustained at T_2_. This intervention has improved inpatient continence care.

## 1. Introduction

Urinary continence issues include urinary incontinence (UI), defined as the involuntary loss of urine [1], and lower urinary tract symptoms (LUTS). LUTS is a term used to describe symptoms associated with urine storage such as urinary retention, bladder outlet obstruction, difficulty initiating a void, and frequency and urgency without incontinence [1]. These symptoms are common for the general population and for inpatients.

Despite up to 43% of adult inpatients experiencing urinary continence issues [2] and international recommendations for optimal urinary continence care [1,3,4,5], there is little reporting of effective interventions to systematically deliver urinary continence care to these inpatients [2,6]. Urinary continence care recommendations include that health services should have systems for assessment, diagnosis of UI/LUTS type, and management that are consistent with best evidence [1,4,5]. The recommendations emphasise shared decision making between clinicians, patients, and their carers. Providing guideline recommended UI/LUTS care is important, not only to minimise the direct effects of UI/LUTS but also to reduce often-associated complications. These include falls [7], urinary tract infections [8], breakdown of skin integrity [9,10], altered mood [11], and bladder overdistension [12]. Although UI/LUTS are often complex, with appropriate inpatient clinical care symptoms can be prevented, managed and even cured, and complications avoided.

As part of our formative research, in 2010 we developed a guideline-based intervention to assist stroke clinicians in three metropolitan inpatient rehabilitation units deliver evidence-based UI/LUTS care [13]. The team synthesised UI/LUTS guideline recommendations into the Stroke Continence Assessment and Management Plan (SCAMP) intervention. SCAMP presented clear, concise, and explicit recommendations for optimal inpatient continence care for people after stroke. The user-friendly intervention guided clinicians through conducting a urinary continence assessment, determining the type of UI/LUTS, and developing an individualised management plan for those with, or at risk of, symptoms. The plan was developed in conjunction with the patient or carer.

Although the details of the SCAMP invention were shared widely at Australian stroke conferences and forums, evidence–practice gaps in continence care for inpatients with stroke continued. Data from Australia in 2017–2018 indicated that a quarter of patients admitted with acute stroke [14] and 41% of inpatients undergoing stroke rehabilitation [15] had urinary incontinence. Of those people with symptoms, 18% in acute [14] and 52% in rehabilitation [15] had a documented urinary continence management plan. Acute stroke and rehabilitation nurses and clinician researchers recognised that the implementation, upscale, and spread of the SCAMP intervention had the potential to improve urinary continence practice not only for stroke but also for other patient populations.

The successful implementation, sustainability, and scalability of interventions is often very complex; however, they can be enhanced by using evidence-based theoretical approaches for implementation [16,17]. Different theoretical approaches can be used for different components of a study, including the design, the systematic planning and development, and the evaluation [16,18,19,20]. Theoretical approaches can be used to identify potential influencers on implementation and to select behaviour change strategies, such as audit and feedback, targeting these influencers [16,18,19,20]. Frameworks and models can assist researchers, managers, and clinicians to integrate best-practice care into practice through behaviour change [17]. To our knowledge, there are no previous studies that are informed by implementation frameworks and models that investigate the feasibility and effectiveness of a practice-change package to improve, then maintain inpatient UI/LUTS care.

The aim of this study was to determine if our practice-change package (implementation of our guideline-based SCAMP urinary continence care intervention) is effective and feasible for improving then maintaining urinary continence care in wards that admit acute and rehabilitation patients with various diagnoses in Australian metropolitan and regional hospitals.

The research questions were:

### 1.1. Primary

Does the implementation of our SCAMP urinary continence care intervention increase the proportion of inpatients with UI/LUTS who have an individually tailored UI/LUTS management plan?

### 1.2. Secondary

1.Does the implementation of our SCAMP urinary continence care intervention increase the proportion of:
(a)Inpatients with UI/LUTS who have an assessment and diagnosis of type(s) of UI/LUTS?(b)Inpatients with UI/LUTS and their caregivers who are involved in the development of the management plan?2.Does the implementation of our SCAMP urinary continence care intervention reduce rates of complications that can be associated with UI/LUTS?3.What is the change in the above outcomes at 12 months after the implementation commenced?4.Is the practice-change package feasible for wards to adopt, with good fidelity to the implementation strategies?

## 2. Materials and Methods

In this paper we report the changes in clinical practice observed at two time points following implementation of our intervention. The study protocol outlines the methods in detail [21]. The study was conducted as described in the protocol and is reported according to the Standards for Reporting Implementation Studies (StaRI) guidelines [22].

### 2.1. Design

We conducted a co-created, pragmatic, before and after implementation study on 15 wards at 12 hospitals in New South Wales and Queensland, Australia. Clinician representatives, predominantly nurses, from each ward were members of the project team from the outset. The study was conducted between December 2018 and February 2022. Inpatient clinicians were the target of the practice-change package. Data were collected via inpatient medical record audits over three 3-month periods: before and after the implementation period and after the maintenance period.

#### Frameworks

To enhance the success of our SCAMP intervention we used evidence-based theoretical approaches for implementation [16]. We used the:Knowledge to Action Framework as the process framework that guided development of the intervention (“knowledge creation” phase) and implementation (“action cycle” phase) [18].Theoretical Domains Framework to identify potential influencers on implementation (barriers and facilitators) and the accompanying COM-B model to identify strategies to address the key barriers [20]. The Theoretical Domains Framework is frequently used when assessing individual-level barriers and facilitators, rather than those at a systems level.RE-AIM Framework (reach (R), effectiveness (E), adoption (A), implementation (I), and maintenance (M)) [19] as it is a useful structure for evaluation implementation efforts. It can be used to evaluate program elements that may improve sustainable adoption and implementation.

The selection of frameworks was part of the co-design process. Clinician researcher members of the team found these highly cited frameworks relatable, as they reflected approaches previously used in ward-based quality activities. The selected frameworks were also felt to be complementary (Theoretical Domains Framework with the COM-B) [20], generalisable (because they had been used for other studies within the Australian context), and applicable to the wards participating in this study.

### 2.2. Sample

#### 2.2.1. Participating Wards

This project was instigated by stroke and rehabilitation clinicians who identified UI/LUTS inpatient care needed to be improved on their ward for people after stroke and potentially for other inpatient populations. In Australia, people after stroke are cared for on wards that admit people with a range of conditions. This care may be provided in a stroke unit embedded on the ward or as part of the general ward population. Fifteen wards at 12 hospitals from four health service districts in Australia participated in this study. The 15 wards were a convenience sample of wards that admit acute and rehabilitation patients with various diagnoses in Australian metropolitan and regional hospitals. To be eligible to participate, key ward clinicians and nurse managers had to identify that UI/LUTS care was an issue for their ward and be willing to commit resources towards improving UI and LUTS care by implementing our SCAMP intervention. The characteristics of each ward are outlined in Table 1.

#### 2.2.2. Target Population

The target population for the SCAMP practice-change package included clinicians, predominantly nurses, working in each participating ward. These clinicians were not trained continence or urology experts. There were no exclusion criteria for clinicians as the study was a service improvement initiative. Neither clinicians nor patients were consented to receive our practice-change package.

#### 2.2.3. Included and Excluded Medical Records

The included adult inpatient populations varied between wards but included acute stroke, acute medicine, and/or rehabilitation for any condition, including stroke (Table 1). Consecutive records of inpatients aged 18 years and older who were discharged from participating wards were included. Patients screened as having no UI/LUTS symptoms or receiving end-of-life care were excluded.

### 2.3. Data Collection

#### 2.3.1. Medical Record Audit

The medical record audit tool used for data collection was developed for this study. It was based on questions in the Australian Stroke Foundation national audits [14,15,23] and the content of the SCAMP decision support tool. The audit tool was piloted, and all data collectors received education before its use.

Data were collected from medical records over three 3-month periods: before implementation (T_0_), immediately after the 6-month implementation period (T_1_), and immediately after the 6-month maintenance period (T_2_). T_0_ data collection for all wards occurred for patients discharged in August–October 2018. The start dates for each ward to commence implementation were staggered due to local competing interests at the time, including the NSW-wide rollout of the electronic medication chart. The first three sites commenced implementation in April 2019. Their T_1_ and T_2_ data collection was for patients discharged in November 2019–January 2020 and in May–July 2020, respectively. The 11th and final ward that completed all data collection undertook T_1_ and T_2_ data collection for patients discharged in February–April 2020 and in August–October 2020, respectively.

The medical records of inpatients were screened for the presence of UI/LUTS symptoms. To reduce selection bias, we screened consecutive records of patients discharged from each ward for each month of each 3-month data collection period. For the excluded records we extracted data for demographic characteristic information, continence status, and how this was determined.

The medical records of inpatients determined to have UI/LUTS were audited. Audits were performed for 15 records for each month or for all patients discharged during that month, whichever occurred first. Audits were conducted at each hospital by the project team members from that hospital and other local clinicians with legitimate access to the records, as per local health service requirements for patient privacy and confidentiality. An online medical record audit data dictionary was available. Information regarding assessment, diagnosis, management plans including the involvement of the patient and carer in the plan, and in-hospital complications were extracted. The in-hospital complications associated with UI/LUTS included were falls [7], urinary tract infections [8], issues with skin integrity [9,10], altered mood [11], and bladder overdistension [12]. Data were extracted into and managed using the REDCap electronic data capture tool [24] hosted on a secure server at the Hunter Medical Research Institute, NSW.

#### 2.3.2. Feasibility and Fidelity Evaluation

To assess the feasibility and the fidelity of the implementation of the intervention on each ward a recording sheet was developed. Project team members from each ward self-reported if and how they adopted an intervention strategy.

### 2.4. Sample Size and Power Calculation

Our primary outcome was the change (T_1_ − T_0_) in the proportion of inpatients who had an individually tailored UI/LUTS management plan. It was determined that 15 consecutive medical record audits per ward per month (i.e., pooled sample of 675 audits anticipated per data collection period) would provide >90% power to detect a 10% absolute increase (from before intervention) in the proportion of symptomatic patients with a UI/LUTS management plan (type 1 error rate of 5%). This calculation conservatively assumed 20% of patients in acute and 50% in rehabilitation wards have a plan before intervention (based on Australian Stroke Foundation national audit results for included wards) [14,15].

### 2.5. Study Intervention—Practice-Change Package

Our practice-change package was the SCAMP intervention that we implemented using theoretically informed implementation strategies. The practice-change package was designed to support inpatient clinicians, predominantly nurses but including educators and managers, to deliver guideline-recommended UI/LUTS care on their ward.

#### 2.5.1. SCAMP Intervention

In 2018 our team revised the three components of SCAMP with nursing, allied health and medical experts from stroke, continence, rehabilitation, and urology to ensure they met current best-evidenced and guideline-recommended UI/LUTS care for most adult inpatient medical and rehabilitation populations. The intervention was renamed the Structured urinary Continence Assessment and Management Plan to reflect the inclusiveness of inpatient populations beyond stroke while retaining the SCAMP acronym.

The SCAMP intervention consisted of:a.The 4-page Structured urinary Continence Assessment and Management Plan (SCAMP) decision support tool, which can be downloaded from within each of the web-based modules below. This tool guides clinicians through conducting a urinary continence assessment, determining the type of UI/LUTS, and developing an individualised management plan for those with or at risk of symptoms in conjunction with the patient or carer.b.The associated Clinical Practice Guideline.c.Eight web-based education modules and a local module on how to use the SCAMP decision support tool (PowerPoint presentation with voice-over). The web-based modules cover information on normal bladder function, why continence is an issue after stroke, and six common inpatient UI and LUTS types. They are hosted on the Stroke Foundation website https://informme.org.au/modules/urinary-continence-and-stroke (accessed on 17 April 2023).

#### 2.5.2. Implementation Strategies

The key barriers identified before implementation and strategies selected with mapping to the COM-B domains are shown in Table 2. Strategies were selected to overcome the barriers identified from three sources. Firstly, we used research of known barriers to clinicians implementing continence guideline recommendations [25]. Secondly, we used the results of our before-implementation clinician questionnaire that was informed by the Theoretical Domains Framework [20], and thirdly the ward-specific barriers that local teams identified using the Behaviour Identification and Mitigation tool [26]. For this tool, local teams asked nursing, allied health, and medical clinicians about the SCAMP decision support tool and guideline. They walked through the process to simulate real-ward circumstances and to identify the barriers to implementation. From their data, each team summarised and prioritised their barriers, then developed a local action plan focused on overcoming the barriers. The practice-change package was adapted by each ward to suit their local context. This included the mode of delivery, dose, and frequency of each local intervention strategy.

### 2.6. Data Analysis

The T_0_, T_1,_ and T_2_ group characteristics and demographics results are presented with descriptive statistics. Categorical data are presented as count (%) or median (interquartile range; IQR) if continuous. All results are presented as aggregated summary measures. Across-period differences in patient characteristics were examined using the Kruskal–Wallis test for continuous variables and the chi-squared test for categorical variables.

Groups were compared with respect to change, from T_0_ to T_1_ and T_2_ using mixed effects logistic regression models. Demographic characteristics that were found to be significantly different across study periods (i.e., Kruskal–Wallis or chi-squared *p* < 0.05) were treated as confounders and included in the regression models as adjusting covariates. This resulted in the mixed effects logistic models having a fixed effect for study period, inpatient age, and inpatient population type, as well as a random intercept for ward. The planned regression analyses were intention-to-treat and included all available data from all wards in all time-points, regardless of participation completeness throughout the study periods. A posteriori per-protocol analyses were also performed, wherein only the included observations from the wards that had complete participation throughout the study was performed to examine the effect of the practice-change package under full uptake conditions. The a posteriori per-protocol analysis was performed using a mixed effects logistic regression with them same model specifications as above. The estimates of each mixed logistic model are presented as odds ratios (OR) with 95% confidence intervals (CI) and p-values. Process data are reported as the proportion of wards that adopted an implementation strategy.

Statistical analyses were programmed using SAS v9.4 (SAS Institute, Cary, NC, USA). A priori, *p* < 0.05 was used to indicate statistical significance.

## 3. Results

### 3.1. Ward Participation

All 15 wards completed the before-implementation audit. The onset of the COVID-19 pandemic in March 2020 affected the conduct of this study, as shown in Table 1. Four wards, including three rehabilitation wards, had to withdraw; two wards were unable to commence implementation; one ward almost completed; and one had just completed implementation before they were closed. Eleven wards contributed data to the after-implementation and maintenance audits.

### 3.2. Characteristics of the Inpatients Whose Medical Records Were Observed

#### 3.2.1. Screening

The medical records of 2298 inpatients were screened for the study. The age of the inpatients screened were consistent over the three data collection periods: median years (Q1, Q3) T_0_ = 78 (68, 86), T_1_ = 76 (65, 84), and T_2_ = 76 (65, 84). For the records screened at each data collection period, approximately 52% were female, 4% identified as Aboriginal or Torres Strait Islander, 70% were in a large city hospital, and 30% were in a regional hospital. The proportion of inpatients with UI/LUTS of those screened during each data collection period were T_0_ = 33% (283/849), T_1_ = 33% (241/740), and T_2_ = 36% (256/709). The proportions with UI/LUTS of those screened varied by patient population: acute stroke T_0_ = 30% (58/191), T_1_ = 24% (44/194), and T_2_ = 23% (44/190); acute medicine T_0_ = 26% (92/359), T_1_ = 33% (125/385), and T_2_ = 33% (113/342); rehabilitation T_0_ = 44% (133/299), T_1_ = 43% (69/161), and T_2_ = 56% (99/177). Inpatients with UI/LUTs compared with those without were 5–8 years older and more likely to be female.

For the inpatients screened as not having UI/LUTS, the method of determining their continence status changed over the three study periods. The proportion who had a documented continence assessment increased from T_0_ = 8% (43/566) to T_1_ = 44% (220/499) and T_2_ = 40% (183/453). For the remainder of the records the auditors had to determine the continence status from the progress notes.

#### 3.2.2. Audits

From the medical record screening, 34% of inpatients were deemed to have UI/LUTS, and their records were audited (inpatients with UI/LUTS = 780: T_0_ = 283, T_1_ = 241, T_2_ = 256). The demographic characteristics for these inpatients, and the statistical significance of the difference in distributions of these characteristics between study periods are shown in Table 3. Patient age at admission and inpatient population type were identified as potential confounders and were included as adjusting covariates in the regression models.

### 3.3. Clinical Care Delivery Outcomes

Changes in clinical practice and the proportions of in-hospital complications associated with UI/LUTS are presented in Table 4, Figure 1, and described below. The results of the intention-to-treat and per protocol analyses were very similar and with identical conclusions; therefore, only the intention-to-treat results are presented.

Implementation of our practice-change package resulted in substantial and statistically significant improvements in continence care. The adjusted odds ratios were approximately 4-fold higher after implementation for receiving a continence assessment [OR (95% CI): 4.4 (2.7–7.0)] or management plan [OR (95% CI): 4.3 (2.3–7.9)]. Receiving a diagnosis of UI/LUTS type(s) was approximately 6-fold higher [OR (95% CI): 6.5 (4.1–10.2)]. The improvements for each of these practice components was sustained from after implementation to the maintenance period.

The proportion of inpatients who received all three components (assessment, diagnosis, and management plan) rose by 18%, from 3% (9/283) before implementation to 21% (44/214) after implementation and 21% (49/232) during the maintenance period. The documented involvement of inpatients or carers in the development of the management plan was low and unchanged across the three timeframes.

### 3.4. Patient Complication Outcomes

There was no statistically significant difference in the proportion of inpatients with complications associated with UI/LUTS across the three study periods (range: falls 7–8%; urinary tract infections 12–17%; issues with skin integrity 2–4%; altered mood 4–9%; and bladder overdistension 5–8%). A third of inpatients in each of the three audit periods experienced one or more complication often related to UI/LUTS (Figure 1).

### 3.5. Feasibility and Fidelity Evaluation

It was feasible for all 13 wards that proceeded to implementation to adopt the practice-change package. The package appears scalable, as the wards included acute (n = 3), rehabilitation (n = 5), and acute and rehabilitation (n = 5) wards in four metropolitan and six regional hospitals. The proportion of wards that adopted each implementation strategy are shown in Table 2. Of the 18 strategies, 12 (67%) were adopted by 100% of wards and 5 (28%) were adopted by 85–92% of wards. Although 92% of the wards started with champions being trained and assigned, this dropped to 31% throughout the implementation phase. This was due to clinicians being on leave, resignations, and changes in personnel in specified roles, such as clinical nurse educators.

## 4. Discussion

In this study we demonstrated a reduction in the evidence–practice gap in UI/LUTS care following the implementation of our SCAMP intervention that targeted inpatient clinicians. After implementation, the proportion of inpatients with UI/LUTS receiving a UI/LUTS assessment, diagnosis of UI/LUTS type(s), and a management plan increased substantially. These improvements were maintained 6-months later, despite the onset of COVID-19 and the subsequent need for wards to make substantial changes in how they functioned. All thirteen wards that completed the implementation phase were able to adopt the SCAMP practice-change package with good fidelity.

Our findings confirm those of other studies that UI/LUTS is commonly experienced by inpatients but there is an evidence–practice gap regarding UI/LUTS care. Across our three medical record audits, 23–30% of acute stroke, 26–33% of acute medical, and 43 to 56% of patients undergoing rehabilitation were deemed to have UI/LUTS. This is in keeping with the studies reviewed by Ostaszkiewicz et al. [2]. These researchers identified that inpatient UI/LUTS prevalence ranged from 11 to 43% across a range of ward types including intensive care, surgical, medical, rehabilitation, and geriatrics. The low proportion of inpatients with UI/LUTS receiving a UI/LUTS assessment (38%), diagnosis of type (30%), and management plans (7%) observed in our before-implementation audit are similar to those found in other studies. In a study by Zurcher et al. (2011), 51% (41/78) of elderly inpatients screened positive for urinary incontinence [27]. However, of these patients with UI, only 24% (10/41) had this documented in their medical record and 5% (2/41) had a documented diagnosis of incontinence type [27]. In a study by Trad et al. (2019), their audit of 100 inpatient medical records for two surgical and two medical wards indicated that 87% of patients had a urinary continence assessment; however, only 14% had a diagnosis of type and 15% received conservative interventions that were tailored to their specific type of incontinence [28].

In the current study we were able to improve, then maintain the proportion of inpatients with UI/LUTS who received three key recommended elements of inpatient urinary continence care. The improvement from before to after implementation in receiving a UI/LUTS assessment (25%), diagnosis of type(s) (40%), and receiving a management plan (17%) are comparatively large improvements for implementation studies. Behaviour change improvements of between 4% and 12% have been reported for multifaceted interventions [29]. The improvements we saw may be due to our multifaceted intervention targeting the identified barriers and facilitators. Education programs alone regarding urinary incontinence for nurses have been shown to improve knowledge but have had mixed effects on attitudes and practice [30].

Our study is one of two studies that we have identified that addresses inpatient UI/LUTS care through theoretically informed implementation and the only one to include a maintenance period. This is despite UI/LUTS being common for inpatients, with well-recognised and considerable evidence–practice gaps in inpatient continence care. This deficit of studies for inpatient UI/LUTS care is reflected in two recent reviews investigating randomised controlled [31,32] and before–after [31] implementation studies of nursing practice. Only two studies addressing urinary continence practice were included in these reviews: one conducted in an outpatient setting [31,32] and one in a nursing home [32]. Similarly to our study, the inpatient continence care implementation study conducted by Trad et al. included a decision tool to guide assessment and management, and education was part of their intervention [28]. These strategies aimed to address their identified barrier of clinician lack of knowledge in continence care. Although the Trad et al. study showed substantial improvement from before to after implementation in assessment (87% to 99%), diagnosis (14% to 75%), and management plans (24 to 74%), there was no subsequent evaluation to determine if these improvements were maintained.

Unfortunately, the proportion of patients with documented involvement of the carer and patient in management planning was low (≤5%) and remained unchanged. This low level of engagement was similar to the study by Trad et al., where no patients received education about incontinence management before implementation and only 5% did after implementation [28]. Although the SCAMP decision tool has a prompt to include the patient and carer in the management plan and to provide them with education, our practice-change package did not include specific training for clinicians for this. The Australian National Safety and Quality Health Service Standards recommend hospitals provide education, training, and resources to equip clinicians to partner with patients in their care [33]. This may be specifically required for UI/LUTS as this can be a sensitive and often taboo topic for patients and clinicians.

The proportion of inpatients experiencing complications did not change, with approximately one third of inpatients experiencing one or more complications that are often associated with UI/LUTS. In our study, urinary tract infections and falls were the most common complications. Urinary tract infections were experienced by 12–17% of inpatients, which is much higher than the 2% who experienced a hospital-acquired urinary tract infection reported by Mitchell et al. [34]. The 8% of patients who had a fall is similar to the 10% identified in general medicine wards in five urban hospitals in Canada [35]. Reducing complications is important to reduce excess morbidity, mortality, and healthcare expenditure. Further investigation is required to determine if our practice-change package can influence complication rates.

Our practice-change package was feasible to adopt for all 13 wards that completed five or more months of the 6-month implementation phase. There was a high level of fidelity for most implementation strategies. This success is likely due to our co-creation approach. From the outset, project leads and champions (predominantly nurses) from each ward, experts in continence and implementation, clinician researchers, and academic nurse researchers were included on the team. End-user members ensured our practice-change package was not only best evidenced but clinically relevant and applicable. Project leads and champions led the practice change on their wards. The lack of a continence nurse within any of the hospitals was recognised from the outset as a barrier to supporting the practice change. This required the upskilling of ward project leads and nurse champions, not only in implementation and conducting the research but also in UI/LUTS care. The monthly virtual team meetings functioned to progress the research and formed a community of practice for the ward leads and champions. The members were able to share their successes, challenges, and locally tailored strategies, in addition to being upskilled in conducting research. Anecdotally, the ward-based project team members reported that although the research was challenging due to time and resource constraints, it was very rewarding to see it succeed and to have the opportunity for personal and professional growth. These themes are similar to those identified by Trad et al. in their inpatient continence care study [28].

### 4.1. Strengths and Limitations

This study has several strengths. We have reported our findings according to the Standards for Reporting Implementation Studies (StaRI) [22] to facilitate replication by other researchers. To determine if any improvements were sustained, we included a maintenance period evaluation. Our practice-change package was tested in metropolitan and regional hospitals and on acute and rehabilitation wards for three patient groups: acute stroke, acute medical, and rehabilitation. This may increase the generalisability and potential scalability of the package. SCAMP may also be applicable to other health conditions and health care settings where providing optimal UI/LUTS care is challenging. To assist other wards improve UI/LUTS care, elements of the intervention (the 4-page Structured urinary Continence Assessment and Management Plan (SCAMP) decision support tool and eight web-based education modules) are freely available on the Stroke Foundation website [https://informme.org.au/modules/urinary-continence-and-stroke (accessed on 17 April 2023)].

There are limitations to this study. We used a before and after design using retrospective medical record audits of consecutive records. A limitation of this study design is that the observed differences cannot be directly attributed to the intervention. Data were extracted by clinicians for their own ward, which has the potential for response bias. However, the use of self-reported, retrospective clinical audits is conventional practice for improvement activities. This potential bias could be mitigated in future research by using blinded assessors. This study was undertaken with only a small amount of research funding (less than AUD 90,000) and required considerable in-kind support from the staff and managers of the participating wards. With this understanding, local clinicians and managers self-selected their ward to participate and the population type to be included in the study. This may limit the generalisability due to potential selection bias. The findings may overestimate the potential effect for wards that admit a low proportion of patients who experience UI/LUTS. Although we describe the implementation strategies and self-reported fidelity, we were not resourced to investigate fully the mode of delivery, dose, and frequencies of each intervention strategy undertaken on each ward. This is a recognised challenge of implementation studies [36].

### 4.2. Recommendations for Further Research

Given the success of this study, further investigations are warranted through larger hybrid design implementation studies using more robust randomised controlled designs, such as step-wedge or cluster randomised controlled trials. These trials could test: the effectiveness by hospital location and type of patient population; the mode of delivery, dose, and frequencies of each intervention strategy; the effect on patient-level outcomes, including continence status, type, co-morbidities, quality of life, and severity of any complications; and scalability and spread. An analysis of the potential economic implications (cost and consequences) for hospitals implementing the SCAMP practice-change package is underway. Our practice-change package was developed from a Western medicine perspective. Further research is required to determine how urinary continence care can be addressed through a cultural lens to ensure we deliver culturally safe and appropriate care for First Nations peoples.

## 5. Conclusions

UI/LUTS is commonly experienced by inpatients, and there is a considerable evidence–practice gap in inpatient continence care. We designed our SCAMP practice-change package for ward clinicians, particularly nurses, who were not trained continence experts to deliver UI/LUTS-guideline-recommended care as part of their usual care. The package was adopted with good fidelity across acute and rehabilitation wards in metropolitan and regional hospitals. Although we improved assessment, diagnosis, and management by what would be considered a good outcome in an intervention study, a large proportion of inpatients still did not receive guideline-recommended care and complication rates did not improve. Further work should be conducted to reduce this evidence–practice gap.

## Figures and Tables

**Figure 1 healthcare-11-01241-f001:**
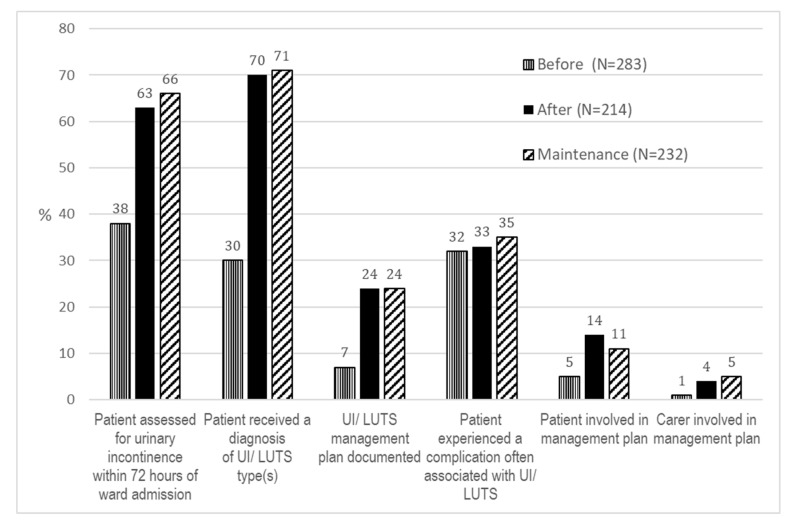
Proportion (%) of inpatients receiving components of UI/LUTS care and experiencing one or more complications often associated with UI/LUTS across the three study periods.

**Table 1 healthcare-11-01241-t001:** Characteristics of participating wards, and the effect of COVID-19 at each phase of the study on each ward.

Hospital/Location	Ward Description	Included Population(s)	Before Implementation Data Collection	Implementation Period (6 Month)	After Implementation Data Collection	Maintenance Period (6 Months) and after Maintenance Data Collection
A Major city	20 bed rehab ward	Rehab	Completed	Completed	Completed	Completed while operating under COVID-19 conditions
	20 bed rehab ward	Rehab	Completed	Completed	Completed	Completed while operating under COVID-19 conditions
	28 bed rehab ward: 20 rehab, 8 neurological. 2 overflow beds	Acute stroke, acute medicine, rehab	Completed	Completed	Completed	Completed while operating under COVID-19 conditions
B Major city	12 bed ward: 8 general medicine, 4 Acute SU	Acute stroke, acute medicine	Completed	Completed	Completed: 1 of 3 months under COVID-19 conditions	Completed while operating under COVID-19 conditions
C Major city	30 bed ward: 26 general medicine, 4 comprehensive SU	Acute stroke, acute medicine	Completed	Completed	Completed: 2 of 3 months under COVID-19 conditions	Completed while operating under COVID-19 conditions
D Regional	32 bed ward: medical and rehab	Acute stroke, rehab	Completed	Completed	Completed: 2 of 3 months under COVID-19 conditions	Completed while operating under COVID-19 conditions
E Major city	32 bed rehab ward	Rehab	Completed	Completed	Study disbanded due to onset of COVID-19 with ward lockdown/closure and furloughing of staff.	
	28 bed general medical ward	Acute stroke, acute medicine	Completed	5 months completed	Study disbanded due to onset of COVID-19 with ward lockdown/closure and furloughing of staff	
F Regional	22 bed rehab ward	Rehab	Completed	Completed	Completed: 1 of 3 months under COVID-19 conditions	Completed while operating under COVID-19 conditions
G Regional	28 bed ward: 24 general medical, 4 Acute SU	Acute stroke	Completed	Completed	Completed: 2 of 3 months under COVID-19 conditions	Completed while operating under COVID-19 conditions
H Regional	16 bed rehab hospital	Rehab	Completed	Completed	Completed: 2 of 3 months under COVID-19 conditions	Completed while operating under COVID-19 conditions
I Regional	28 bed ward: 4 Acute SU, 8 MAU, 16 respiratory/cardiac	Acute stroke	Completed	Completed	Completed: 3 of 3 months under COVID-19 conditions	Completed while operating under COVID-19 conditions
J Regional	24 bed ward: 20 general rehab, 4 comprehensive SU	Acute stroke	Completed	Completed	Completed: 1 of 3 months under COVID-19 conditions	Completed while operating under COVID-19 conditions
K Regional	18 bed hospital: 8 rehab, 10 general medical	Rehab	Completed	Study disbanded due to onset of COVID-19		
L Regional	16 bed rehab ward	Rehab	Completed	Study disbanded due to onset of COVID-19		

SU = stroke unit, Rehab = rehabilitation, MAU = Medical Assessment Unit.

**Table 2 healthcare-11-01241-t002:** The key identified barriers and implementation strategies mapped to the COM-B model and the proportion of the 13 wards that reached the implementation phase that adopted the strategy.

Key Barriers	COM-B Model	Strategy	Strategy Adopted by Ward [n/13 (%)]
Ward leads and champions have limited knowledge and experience in conducting implementation projects	Capability—Physical and Psychological	Identify and prepare—2 implementation workshops conducted	13 (100%)
	Motivation—Reflective and Automatic	Monthly virtual community of practice meetings, out of session phone calls, and emails with project leads	13 (100%)
- UI/LUTS usually a comorbidity, not the main reason for admission so may be overlooked - Clinicians not aware of UI/LUTS evidence–practice gap - Need to change local processes to adopt formalised UI/LUTS care	Motivation—Reflective Capability—Psychological	Audit and feedback of before-implementation results to raise awareness/highlight evidence–practice gap—via ward meetings, emails	13 (100%)
	Motivation—Reflective	Conduct BIM tool to identify local barriers and facilitators	13 (100%)
	Opportunity—Physical	Develop local action plan	12 (92%)
	Motivation—Reflective Opportunity—Social	Audit and feedback—spot check audits to determine what part of the process performed well and by who and what can be improved. Feedback via safety huddles, ward meetings, emails	13 (100%)
	Opportunity—Physical	SCAMP decision support tool embedded into routine practice	13 (100%)
	Capability—Physical	Intensive education and upskilling phase to achieve a critical mass prior to launch	13 (100%)
	Motivation—Automatic	Launch/promotional activities	12 (92%)
- Ward clinicians are not experts in continence, with no/little access to community-based continence nurses - Clinicians perceive they lack knowledge, skills, and confidence in continence care, particularly diagnosis and management plans	Capability—Psychological	Education (meetings, web-based modules) to increase knowledge UI/LUTS types, using SCAMP tool	13 (100%)
	Capability—Physical	Upskilling 1:1 with ward champion/lead	13 (100%)
	Capability—PhysicalMotivation—Reflective	Local champions identified and trained as resource people	12 (92%)
	Opportunity—Physical	Local champions available throughout implementation	4 (31%)
	Motivation—Automatic	Recognition from manager/local project lead of individual staff who did well	11 (85%)
Clinicians need to remember to use SCAMP tool	Capability—Psychological	Written reminders—including posters displayed in ward/emails/SCAMP resource folder	13 (100%)
	Opportunity—Physical	Verbal reminders—including safety huddles/1:1s/ward meetings	13 (100%)
Maintaining improvements	Capability—Psychological	SCAMP education embedded into onboarding of new nursing staff	13 (100%)
	Motivation—Reflective	Spot check audit and feedback—via safety huddles, ward meetings, emails	10/11 (91%) *

* Strategy used during maintenance phase, therefore no data for the two wards that disbanded the study after the implementation phase due to the effect of the onset of COVID-19 on the two wards (Table 1).

**Table 3 healthcare-11-01241-t003:** Demographic characteristics of patients deemed to have UI/LUTS across study periods and indicating any significant difference in the distribution of the characteristic across study periods.

Demographic Characteristic		Before Implementation (n = 283)	After Implementation (n = 214)	Maintenance (n = 232)	*p*-Value
Age at admission (years)	Median (Q1, Q3)	83 (72, 88)	81 (69, 87)	78 (70, 85)	0.004 **
Age group	18–64	36 (13%)	37 (17%)	40 (17%)	0.020 **
	65–74	47 (17%)	42 (20%)	46 (20%)	
	75–84	83 (29%)	65 (30%)	86 (37%)	
	85+	117 (41%)	70 (33%)	60 (26%)	
Sex	Female	161 (57%)	119 (56%)	134 (58%)	0.631
	Male	122 (43%)	93 (44%)	97 (42%)	
	Other	0	1 (0.5%)	0	
Indigenous status *	Indigenous	3 (1.1%)	9 (4.2%)	9 (3.9%)	0.065
Location of hospital	Large city	199 (70%)	164 (77%)	163 (70%)	0.220
	Regional	84 (30%)	50 (23%)	69 (30%)	
Patient population	Acute stroke	58 (20%)	41 (19%)	38 (16%)	0.003 **
	Acute Medical	92 (33%)	104 (49%)	95 (41%)	
	Rehabilitation	133 (47%)	69 (32%)	99 (43%)	

* Indigenous status: inpatient self-identified as Aboriginal and/or Torres Strait Islander. ** *p <* 0.05.

**Table 4 healthcare-11-01241-t004:** Adjusted mixed logistic regression results for aspects of care and complications: intention-to-treat analyses.

		Intention-to-Treat Analysis *		
Outcome	Study Period Comparison	OR (95% CI)	*p*-Value	N in Model
Inpatient assessed for UI/LUTS	After implementation vs. Before implementation	4.38 (2.73, 7.03)	<0.001	721
	Maintenance vs. Before implementation	4.70 (2.94, 7.52)	<0.001	
	Maintenance vs. After implementation	1.07 (0.70, 1.65)	0.745	
Inpatient received diagnosis of UI/LUTS type	After implementation vs. Before implementation	6.49 (4.13, 10.20)	<0.001	729
	Maintenance vs. Before implementation	6.01 (3.82, 9.48)	<0.001	
	Maintenance vs. After implementation	0.93 (0.60, 1.42)	0.726	
Inpatient received UI/LUTS management plan	After implementation vs. Before implementation	4.29 (2.32, 7.94)	<0.001	712
	Maintenance vs. Before implementation	4.03 (2.16, 7.50)	<0.001	
	Maintenance vs. After implementation	0.94 (0.59, 1.49)	0.788	
In-hospital complication often associated with UI/LUTS	After implementation vs. Before implementation	1.42 (0.93, 2.16)	0.106	729
	Maintenance vs. Before implementation	1.48 (0.98, 2.22)	0.061	
	Maintenance vs. After implementation	1.04 (0.69, 1.57)	0.841	
Inpatient involved in the development of UI/LUTS management plan	After implementation vs. Before implementation	0.95 (0.29, 3.08)	0.925	127
	Maintenance vs. Before implementation	0.48 (0.15, 1.57)	0.224	
	Maintenance vs. After implementation	0.51 (0.21, 1.21)	0.125	
Carer involved in the development of management plan	After implementation vs. Before implementation	1.41 (0.32, 6.22)	0.646	127
	Maintenance vs. Before implementation	1.45 (0.34, 6.22)	0.612	
	Maintenance vs. After implementation	1.03 (0.37, 2.84)	0.956	

* Adjusted for patient age and patient population (acute stroke, acute medicine, or rehabilitation).

## Data Availability

The data presented in this study are available on request from the corresponding author. The data are not publicly available as they were collected via medical record audit of data routinely collected in hospital. Patients whose records were audited have not consented to data sharing.

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
