# Peer review of "Improving Practice for Urinary Continence Care on Adult Acute Medical and Rehabilitation Wards: A Multi-Site, Co-Created Implementation Study"

_healthcare, 2023, doi:10.3390/healthcare11091241_

Round 1
Reviewer 1 Report
First and foremost, thank you for submitting your research to the journal. Additionally, I would like to applaud you on your novel research with the aim of improving inpatient care in an everyday practice aspect seen particularly in rehab patients. A few suggestions to improve your paper:
1. Please edit to improve the scientific language and tone of the paper.
Some suggestions (not exhaustive) on how this can be improved: For example, in the introduction, consider compounding some sentences and using words and phrases such as "defined as"... e.g., maybe consider rephrasing the second sentence to read "UI is defined as the voluntary loss of urine." or even combining the first and second sentence.
Avoid using "we" if you cite your own research or when stating what was done in this research.
Additionally, when referring to studies later in the discussion, consider using phrases such as "in the study conducted by FirstAuthor et al..." rather than "in one study" and "in another study."
2. In regard to formatting: please put commas between references if there is more than one cited.
3. In the Materials and Methods Section, please clearly define inclusion and exclusion criteria. In addition, I would delete the portion where each author is mentioned as being part of the team since this would be understandable that they participated in the research given their authorship. If you would like to mention their roles, this would probably be more appropriate to do so later in the author contribution or acknowledgment section.
4. Abbreviations should only be defined once and then used throughout consistently without needing to define them again (e.g., T0).
5. Consider improving the data analysis section (i.e., software used, what analysis was used for which part, etc.)
6. Discussion could be improved. Rather than stating there are several strengths and that there are limitations to the study, consider presenting this a little differently and improving the scientific language rather than merely stating.
Overall, the research is extremely valuable and provides insight into how we can change inpatient urinary continence care moving forward. After this article is revised and improved, I would strongly recommend this for publication.
Reviewer 2 Report
It would be interesting that the authors specified in the methods the range of months of follow-up in the period they call T2, instead of using an ambiguous date such as "after the 6-month maintenance period".
It would be interesting that the authors further describe how they have come to determine these frameworks rather than others.
For example in the Barriers and facilitators.
In the descriptive table it can be seen that there is a high percentage of older patients, so it would be convenient to expand the descriptive part with variables that can be confounding factors: disability, comorbidity, frailty, anticholinergic drugs, neurological diseases... since these factors are frequent in people with urinary incontinence and are risk factors for consequences such as falls, and in the descriptive analysis it can be observed that there are statistically significant differences in some variables such as age (being more frequent in this group in addition to these factors).
For the rest I consider that the study has a very well structured methodology, it is interesting and its exposition is adequate. But these points would be important to truly demonstrate that the results may be due to the implementation of the protocol and not to differences in the characteristics of the sample.
Reviewer 3 Report
Congratula on the manuscript. I will provide you with some comments and suggestions for the authors:
Introduction:
The introduction provides sufficient background and includes all relevant references.
References:
All the cited references are relevant to the research.
Research Design:
The research design is appropriate, but a quasi-experimental before-and-after design with retrospective medical record audits has limitations. A hybrid design implementation study with a randomized controlled design, such as a step-wedge or cluster randomized controlled trial, could be conducted to improve the study design (future studies suggestion).
Methods:
The methods are adequately described, but the authors could provide more information on the mode of delivery, dose, and frequency of each intervention strategy on each ward.
Results:
The results are presented, but the authors could provide more information on the effect of the interventions on patient-level outcomes, including continence status, type and severity of any complications.
Discussion:
Address the limitations of the study in more detail and suggest potential solutions or areas for further research.
Example: The authors could provide more detail on the potential sources of bias in the study, such as the use of self-reported retrospective clinical audits. They could suggest ways to mitigate these biases in future research, such as using objective measures or blinded assessors. Additionally, the authors could provide more detailed recommendations for further research, such as exploring the cultural safety of the practice-change package for First Nations peoples or investigating the effectiveness of different modes of delivery, dose, and frequency of intervention strategies.
Emphasize the practical implications of the study's findings for inpatient continence care.
Example: The authors could provide more specific recommendations for how the practice-change package could be implemented in other healthcare settings to improve inpatient continence care. They could also highlight the potential cost-effectiveness of the package and the potential benefits for patients, such as improved quality of life and reduced risk of complications.
Provide a more nuanced analysis of why certain intervention strategies were more effective than others.
Example: The authors could provide more detailed information on the fidelity of the intervention strategies and whether certain strategies were more difficult to implement or required more resources. They could also analyze whether certain patient populations or types of hospitals responded better to certain strategies. Additionally, they could discuss any unexpected or unintended consequences of the intervention strategies and provide insights into how these could be addressed in future implementation efforts.
Conclusions:
The results support the conclusions, but many inpatients still did not receive guideline-recommended care, and complication rates did not improve. Further work should be conducted to reduce this evidence-practice gap.
Originality / Novelty:
The study is not particularly original or novel, but it addresses an important evidence-practice gap.
Significance of Content:
The study is significant because it addresses an important evidence-practice gap in inpatient continence care.
Quality of Presentation:
The presentation quality is good, but the authors could provide more details on the mode of delivery, dose, and frequencies of each intervention strategy on each ward.
Scientific Soundness:
The scientific soundness is good, but a hybrid design implementation study with a randomized controlled design could improve the study design (for a future study).
Interest to the Readers:
The study is likely to be of interest to readers who are interested in evidence-practice gaps in inpatient continence care.
Comments and Suggestions for Authors:
Overall, the authors have presented a well-designed study that addresses an important evidence-practice gap in inpatient continence care. To improve the study design, the authors could conduct a hybrid design implementation study with a randomized controlled design, such as a step-wedge or cluster randomized controlled trial. In addition, the authors could provide more information on the mode of delivery, dose, and frequencies of each intervention strategy on each ward, as well as more information on the effect of the interventions on patient-level outcomes, including continence status, type and severity of any complications.
Round 2
Reviewer 2 Report
Congratulations, I think that the paper could be publish in present from.